# Two-Stream Network One-Class Classification Model for Defect Inspections

**DOI:** 10.3390/s23125768

**Published:** 2023-06-20

**Authors:** Seunghun Lee, Chenglong Luo, Sungkwan Lee, Hoeryong Jung

**Affiliations:** 1Division of Mechanical and Aerospace Engineering, Konkuk University, 120 Neungdong-ro, Gwangjin-gu, Seoul 05029, Republic of Korea; erioer95@konkuk.ac.kr (S.L.); luo0611@konkuk.ac.kr (C.L.); 2Sambo Technology, 90 Centum Jungang-ro, Haeundae-gu, Busan 48059, Republic of Korea; sales@sambotechnology.co.kr

**Keywords:** defect inspection, machine vision, one-class classification, two-stream network

## Abstract

Defect inspection is important to ensure consistent quality and efficiency in industrial manufacturing. Recently, machine vision systems integrating artificial intelligence (AI)-based inspection algorithms have exhibited promising performance in various applications, but practically, they often suffer from data imbalance. This paper proposes a defect inspection method using a one-class classification (OCC) model to deal with imbalanced datasets. A two-stream network architecture consisting of global and local feature extractor networks is presented, which can alleviate the representation collapse problem of OCC. By combining an object-oriented invariant feature vector with a training-data-oriented local feature vector, the proposed two-stream network model prevents the decision boundary from collapsing to the training dataset and obtains an appropriate decision boundary. The performance of the proposed model is demonstrated in the practical application of automotive-airbag bracket-welding defect inspection. The effects of the classification layer and two-stream network architecture on the overall inspection accuracy were clarified by using image samples collected in a controlled laboratory environment and from a production site. The results are compared with those of a previous classification model, demonstrating that the proposed model can improve the accuracy, precision, and F1 score by up to 8.19%, 10.74%, and 4.02%, respectively.

## 1. Introduction

Defect inspection is important in the manufacturing industry and is required to ensure consistent product quality and improve the costs and efficiency of the entire manufacturing process. Human visual inspection, however, is time-consuming, labor-intensive, and prone to human errors. In contrast, machine vision inspection using cameras, optics, and inspection software enables fast and robust low-cost inspection. Therefore, it has been increasingly adopted in various manufacturing industries [1,2,3,4,5]. For decades, numerous studies on machine vision inspection have been conducted [6,7,8,9,10], but traditional inspection techniques still face challenges in dealing with variations in environmental conditions and part appearance.

Recently, inspection algorithms integrating artificial intelligence (AI) techniques have shown promise and improved the accuracy and robustness of defect inspection. These algorithms have been employed in various manufacturing industries, including textile [11], fabric [8,9,10], and steel surface [4,12]. Defect inspection using deep learning algorithms achieved enhanced accuracy and robustness by learning features from the large training dataset. A number of prominent architectures and pre-trained models, such as AlexNet [13], VGGNet [14], ResNet [15], and MobileNet [16], have emerged, and these are accompanied by various techniques to enhance inspection performance. Wei et al. achieved an inspection accuracy of 98.5% using convolutional neural network (CNN)-based algorithms with image preprocessing, such as noise reduction and binarization, to detect defective products in the textile industry [17]. Yang et al. used the you only look once (YOLO) v5 object detection algorithm to detect and identify welding defects on steel pipes. The proposed model achieved an accuracy of 97.8%, demonstrating its potential for real-time welding defect detection [18]. Kim et al. presented a skip-connected convolutional autoencoder for advanced printed circuit board (PCB) inspection. The proposed unsupervised autoencoder model delivered promising performance, with a detection rate of up to 98% in 3900 defect and non-defect images [19]. Tang et al. proposed a skip autoencoder to improve the accuracy of anomaly detection and address labeling issues. Leveraging a pre-trained feature extractor and skip connections, the proposed method achieved better performance, showing a maximum area under the curve (AUC) of 0.98 [20]. Upadhyay et al. developed a U-Net-based deep learning framework to detect engine defects. They applied a hybrid motion deblurring method for image sharpening and denoising, combined with a customized generative adversarial network (GAN) model, to remove the blur effect based on classic computer vision techniques. The deep learning framework achieved precisions and recalls of over 90% [21]. Yoon Jong-Pil et al. presented a defect classification approach based on a convolutional variational autoencoder (CVAE) and deep CNN for metal surface defect inspection. The proposed conditional CVAE achieved a maximum completion of 0.9969 [22].

Although AI-based inspection provides superior performance compared to traditional methods, several limitations remain in applying this approach to practical situations. One major challenge is the performance degradation caused by data imbalance. AI-based inspection requires a large training dataset. However, practically, the collected data often suffer from class imbalance, where certain classes have considerably fewer samples than others. In defect inspection, collecting sufficient defective samples is difficult because the defect rate is quite low (usually under 1–5%) in general manufacturing processes. To address this issue, various methods have been proposed, including data augmentation [23,24,25], synthetization [19,26], and an adjustment of the weight or loss function of the network [27]. Wang et al. proposed a novel loss function called ‘mean false error’ together with its improved version called ‘mean squared false error’ for deep network training using imbalanced datasets [28]. Mao et al. improved data imbalance by extending the training dataset using a GAN model and achieved up to 86.8% accuracy [29].

One-class classification (OCC), which identifies objects belonging to a specific class given only positive samples of that class, is attracting attention as a solution to this problem [30,31,32,33,34,35,36,37,38,39,40]. Unlike general machine-learning-based classification algorithms, the OCC model aims to learn a classification boundary that separates the target class from other classes in the input space. OCC can thus be utilized effectively to solve data imbalance problems, as it does not require negative samples and can be trained only using positive samples. Shin et al. proposed a one-class support vector machine (SVM) model to detect mechanical defects in electronic devices, achieving up to 93.9% accuracy compared to the multilayer perceptron method [31]. Ruff et al. proposed a deep support vector data description that extracts the similarity between patterns of general categories and new data. The proposed method achieved up to 99.7% average AUCs on MNIST and CIFAR-10 [34]. Lee et al. proposed a one-class deep-learning-based fault-detection module for imbalanced industrial time-series data. Using four different networks, i.e., MLP, ResNet, LSTM, and ResNet-LSTM, for prediction, they achieved an excellent fault prediction accuracy of 96% [36]. Goyal et al. developed a deep robust one-class classification (DROCC) to help address the representation collapse problem. The DROCC achieved an average accuracy of 74.2% using the CIFAR-10 dataset [37].

The representation collapse problem is a major issue in OCC, and it can arise when the diversity of the training data is insufficient, or the data follow a repetitive pattern. In such cases, the decision boundary is fitted too tightly to the training dataset, leading to a decrease in the generalization performance for new data. In practical applications, the environmental conditions for collecting training and test samples may not be the same, which can lead to false positive errors, resulting in overall performance degradation.

In this paper, we propose a two-stream network OCC model for defect detection that attempts to address the representation collapse problem, which has been a critical issue when applying the OCC model to practical applications. The proposed two-stream network model alleviates the representation collapse problem by introducing two feature extractor networks, i.e., global and local feature extractor networks. The global feature extractor network, which is designed to learn a general feature of the target class, can extract a feature vector that is not affected by variations in environmental conditions. The local feature extractor network is designed to capture features specific to the training dataset, and it extracts the target class-oriented feature vectors. Two feature vectors output from each network are merged and passed through the following classification layer for the final decision. Three types of classification layers, i.e., a one-dimensional (1D) convolution layer, a fully connected layer, and an SVM layer, were tested for the target class classification to determine the optimal classification layer. The proposed two-stream OCC model was verified by using an image dataset obtained from the practical application of automotive airbag bracket inspection. The main contributions of this paper are as follows:A two-stream network architecture composed of global and local feature extractor networks is proposed to resolve the representation collapse problem of the OCC model.The classification performances of three types of classification layers, i.e., 1D convolution, fully connected, and SVM layers, are described to elucidate the type that yields the optimal classification performance.The performance of the proposed OCC model is verified using the practical application of automotive airbag bracket inspection.

## 2. Materials and Methods

### 2.1. Two-Stream Network OCC Model

OCC involves training a model using data from a single class and capturing its feature vectors. Although OCC is effective in capturing the distribution of given target data, its ability to recognize new data with different characteristic distributions may be diminished. To address this limitation, which is called representation collapse, this paper proposes a two-stream network OCC model. The main idea is to introduce a global feature extractor network to alleviate the issue of decision boundary collapse relative to the training data. By merging a global feature vector representing object-oriented general characteristics with a class-oriented local feature vector, the two-stream network model prevents the decision boundary from being overfitted to the training data and balances both features to identify an appropriate decision boundary.

Figure 1 shows the two-stream network OCC model proposed in this paper. It consists of two types of feature extractor networks, i.e., global and local feature extractor networks. The global feature extractor network is designed to capture all characteristics of the inspection objects, such as geometrical and topological characteristics. Generally, the global feature is an object-oriented characteristic, and it can be consistently extracted regardless of variations in environmental conditions. The local feature extractor is responsible for extracting the target class-oriented characteristics from the training datasets. The local feature describes the surface characteristics of inspection objects, such as colors and textures. Unlike the global feature, the local feature presented in the image can be influenced by environmental conditions. The two feature vectors obtained from each feature extractor network are merged as a single feature vector and passed through the classification layer.

The global feature extractor network is implemented using an Inception V3 network model that consists of a deep neural network architecture including 94 convolution layers and 20,861,480 parameters. It includes three inception modules, which are composed of multiple parallel paths with different filter sizes, to create a rich set of features that capture different aspects of the input image. The inception modules and auxiliary classifiers in the global feature extractor network alleviate the overfitting problem and improve the consistency of feature extraction. The details of the global feature extractor network are presented in Table 1. The global feature extractor network is pre-trained using an ImageNet dataset separately from the other parts of the entire two-stream network. In the entire model training process, the weights of the global feature extractor network are fixed as the pre-trained value to prevent the feature vector from being biased relative to the training dataset. The global feature vector, $F_{g}$, extracted from the global feature extractor network can be expressed as
(1)$$F_{g}=K_{g}*I,\;F_{g}\in R^{D},$$
where I represents the image, $K_{g}$ denotes the global feature extractor network, and D is the dimension of the global feature vector.

The local feature extractor network is composed of four convolution layers and three max-pooling layers including 3,796,480 parameters as presented in Table 2. A simple CNN structure is used for the local feature extractor network to capture the features specific to the target dataset. The local feature extractor network captures the target data-oriented local feature vector $F_{l}$, which can be determined by applying
(2)$$F_{l}=K_{l}*I,\;F_{l}\in R^{D},$$
where I represents the image, and $K_{g}$ denotes the local feature extractor network. The dimension of the local feature vector is identical to that of the global feature vector. The global and local feature vectors are merged as a single feature vector, $F_{u}$, as follows and passed through the classification layer:(3)$$F_{u}=F_{g}⊕F_{l},\;F_{u}\in R^{2D}$$
where $F_{u}$ is the unified feature vector. $F_{u}$ is passed through the classification layer to determine the final decision of the defect inspection. Three types of classification layers, including a 1D convolution layer, a fully connected layer, and an SVM layer, were implemented to validate the effect of the classification layer on the overall inspection performance and to identify the optimal classification layer. The details of each classification layer are presented in Table 3.

### 2.2. Model Verification

The two-stream network OCC model was verified using the image samples collected by the practical vision inspection system of an automotive airbag bracket. The airbag bracket was manufactured using projection welding, joining a nut on a bracket plate. Faults may have occurred in the welding procedure, resulting in several types of defects such as nut omissions, axial twisting, and surface abnormalities, as shown in Figure 2. These types of defects should be detected by the vision inspection system, and this study verifies the performance of the proposed two-stream network OCC model by evaluating the inspection accuracy using positive and negative airbag bracket samples.

#### 2.2.1. Data Collection

The image datasets for training and performance evaluation were collected in two different environments, i.e., a laboratory and a production site. The vision system, including the camera, lens, lighting, and kinematic configuration, was set identically in both environments, as shown in Figure 3a,b. An area scan monocamera (acA2440-20gm, Basler, Ahrensburg, Germany) with a resolution of 2448×2048 (5 MP) and a 16 mm lens (MVL-KF1628M, HIKROBOT, Zhejiang, China) was used as the vision system. The working distance between the lens and the airbag bracket was set to 10.0 cm. A total of 870 images of airbag bracket samples, including 696 positive and 174 negative images, were collected in the laboratory setup, and 136 images, including 122 positive and 14 negative images, were captured in the production site setup. Subsequently, 80% of the images collected in the laboratory were used to train the two-stream network model, and the remaining 20% were used for model verification. The images collected on the production site were used only for model verification. Figure 4 and Figure 5 show the airbag bracket image samples collected in the laboratory and on the production site, respectively.

#### 2.2.2. Training

The region of interest (ROI) for the airbag bracket’s inspection can be defined as the rectangle centered at the bracket’s center that tightly encloses the nut region. The ROI was cropped in raw image samples and resized to 750×750 for model training and verification. To enlarge the training dataset, several variations were applied to the raw images: The center of the cropped region was randomly set within 100 pixels at the center of the bracket to reflect possible variations in the bracket position, and each image was rotated by 90°, 180°, and 270° and flipped. A total of 3470 image samples were used for training. Figure 6 shows the dataset enlargement procedure applied for model training. The Adam optimizer and Huber loss function were used for training, and the maximum number of epochs was set to 100.

#### 2.2.3. Evaluation Metrics

The performance of the proposed two-stream network model was evaluated by four metrics: accuracy, precision, recall, and F1 score. These evaluation metrics can be determined as follows:$$Accuracy=\frac{TP+TN}{TP+TN+FP+FN},\;Precision=\frac{TP}{TP+FP}$$
(4)$$Recall=\frac{TP}{TP+FN},\;F1score=2\cdot \frac{Precision\cdot recall}{Precision+recall}$$
where TP, TN, FP, and FN represent the true positive, true negative, false positive, and false negative, respectively.

## 3. Results

The performance of the proposed two-stream network model was evaluated from three perspectives: the effect of the classification layer, the effect of the two-stream network architecture, and performance in comparison with those of previous methods. In the performance evaluation, the two-stream model was trained only with the datasets gathered in the laboratory, and it was tested using two datasets gathered in the laboratory and on the production site.

### 3.1. Performance Evaluation in Terms of the Classification Layer

The proposed two-stream network OCC model was implemented with three types of classification layers: 1D convolution, fully connected, and SVM layers. Figure 7 and Table 4 present the experimental results of the two-stream network model according to the selected classification layer, as evaluated using laboratory datasets. In total, 140 positive and 34 negative images collected in the laboratory were used in this experiment. The confusion matrices in Figure 6 demonstrate that the SVM and 1D convolution layers achieved the best performance in classifying the TP (136/140) and TN (23/34) labels, respectively. The 1D convolution layer showed the best accuracy, precision, and F1 score of 0.8966, 0.9236, and 0.9366, respectively, whereas the SVM layer yielded the highest recall of 0.9714, as shown in Table 4.

Figure 8 shows the experimental results, as evaluated by using the production site dataset. In total, 122 positive and 14 negative images collected on the production site were used in this experiment. The confusion matrices in Figure 8 demonstrate that the fully connected and 1D convolution layers achieved the best performance in classifying the TP (119/122) and TN (14/14) labels, respectively. The 1D convolution layer showed the best accuracy, precision, and F1 score of 0.9706, 1.0000, and 0.9833, respectively, whereas the fully connected layer achieved the highest recall of 0.9754, as shown in Table 4.

### 3.2. Performance Evaluation of the Two-Stream Network Model

The performance of the two-stream network model was compared with those of models without one of the global and local feature extractor networks in this experiment. The 1D convolution layer was used for the classification layer in this experiment. Table 5 presents a comparison of the performance of the two-stream network model with those of the single-stream network. The performance evaluation was conducted for both the laboratory and production site datasets. The global feature extractor network model exhibited the lowest performance for both datasets, with an accuracy of 0.8621, a precision of 0.8580, and an F1 score of 0.9205 for the laboratory dataset; and an accuracy of 0.8971, a precision of 0.9030, and an F1 score of 0.9453 for the production site dataset. The 2S-1DOC model exhibited the highest performance for both datasets, with an accuracy, precision, and F1 score of 0.8966, 0.9236, and 0.9366, respectively, for the laboratory dataset; and an accuracy, precision, and F1 score of 0.9706, 1.0000, and 0.9833, respectively, for the production site dataset. Figure 9 presents the t-distributed stochastic neighbor embedding (t-SNE) plots of the feature vectors output from the local, global, and two-stream network. The t-SNE plot visualizes the similarity between feature vectors by mapping high-dimensional feature vectors to a lower-dimensional space (2D). The feature vectors of the two-stream network, which combines the characteristics of the local and global feature extractor networks, clearly distinguish the true and false samples with a single decision boundary.

### 3.3. Performance Comparison with Previous Models

Table 6 compares the performance of the two-stream network model and previous image classification models. Six representative classification models, InceptionV3 [41], ResNet101V2 [14], Xception [42], MobileNetV2 [15], VGG-16 [13], and PaDiM [43], were tested for the performance comparison. The two-stream network model presented the highest accuracy and precision of 0.8966 and 0.9236, respectively. However, ResNet101V2, Xception, MobileNetV2, and VGG-16 yielded the highest recall result of 1.000, and PaDiM shows the highest F1 score result of 0.9388. The InceptionV3 model presented the lowest accuracy, precision, and F1 scores of 0.8621, 0.8580, and 0.9205, respectively. Figure 10 presents the t-SNE plots of the feature vectors of the two-stream network model and previous models. As shown in the figure, the proposed two-stream network most clearly distinguished the true and false samples compared to previous models.

Table 7 presents a comparison of the results obtained using the proposed and previous models and the production site dataset. The two-stream network model presents the highest accuracy, precision, and F1 scores of 0.9706, 1.0000, and 0.9833, respectively.

## 4. Discussion

In the manufacturing sector, defect inspection using AI technology has been extensively studied to optimize labor costs and process automation. However, due to the difficulty of collecting data in the field and data imbalances, OCC has recently attracted attention for various applications. OCC is efficient in applications where data are unbalanced, but it has a critical limitation in that the features are compressed in the training data, resulting in false-positive errors. To overcome this limitation, we developed a two-stream network OCC model consisting of local and global feature extractor networks followed by a classification layer. The performance of the proposed model was validated using a practical example of automotive-airbag bracket-welding defect inspection. The image datasets of the airbag bracket collected in two different environments, i.e., a laboratory and a production site, were used for the training and validation of the proposed model. For the dataset collected in the laboratory, our model achieved results of 0.8966, 0.9236, 0.9500, and 0.9366 for the accuracy, precision, recall, and F1 score, respectively. For the production site dataset, the model achieved results of 0.9706, 1.0000, 0.9672, and 0.9833 for the accuracy, precision, recall, and F1 score, respectively.

The inspection performance of the entire model could be affected by not only the performance of the feature extraction layer but also that of the classification layer. Three types of classification layers, 1D convolution, fully connected, and SVM layers, were tested to investigate the effect of the classification layer and to identify the optimal classification presenting the best inspection performance. The 1D convolution layer showed the best accuracy, precision, and F1 score for both laboratory and production site datasets. The fully connected layer yielded slightly better performances than the 1D convolution layer only in terms of recall. In the performance comparison between the laboratory and production site datasets, the SVM layer exhibited a decrease in the accuracy, precision, recall, and F1 score by 9.44%, 0.24%, 8.87%, and 4.58%, respectively, for the production site dataset compared with the laboratory dataset. By contrast, the 1D convolution layer showed an increase of 8.37% in accuracy, 8.54% in precision, 1.77% in recall, and 5.01% in the F1 score for the production site dataset compared to the laboratory dataset. These results indicate that the classification by the 1D convolution layer is more appropriate for alleviating the representation’s collapse than that by other layers.

Compared with the single-stream network model, the two-stream network model showed an increase of up to 7.35% in accuracy, 9.70% in precision, and 3.80% in F1 score, proving that the two-stream model achieved a better performance than the existing single-stream model. In addition, the proposed two-stream model exhibited performance improvements in the production site dataset’s results compared with the laboratory dataset results, with an increase in accuracy of 8.25%, precision of 8.27%, recall of 1.81%, and F1 score of 4.99%, demonstrating that the proposed model maintains the inspection performance for the datasets gathered under different environmental conditions than the training datasets. This finding proves that the two-stream network architecture contributes to reducing the performance degradation caused by representation collapse.

The effect of the two-stream network on performance improvement is clearly presented by the t-SNE plots shown in Figure 9. In Figure 9a, the feature vectors produced by the global feature extractor network provide a rough classification of the true and false samples, and there is some overlap observed among certain portions of the samples. The lack of a distinct decision boundary can be attributed to the global feature extractor network’s emphasis on capturing general features. In contrast, the feature vector generated by the local feature extractor network depicted in Figure 9b exhibits clear differentiation between true and false samples. Nevertheless, determining a single decision boundary is challenging as false samples are divided into two separate clusters. By combining the characteristics of the global and local feature extractor networks, the feature vector generated by the two-stream network depicted in Figure 9c effectively discriminates between true and false samples using a single decision boundary.

The comparison between the proposed two-stream network model and the previous model confirmed its enhanced classification performance. In the performance comparison with the previous model, the proposed two-stream model showed the best performance for most performance indices, including the accuracy, precision, and F1 score for production site datasets. The improvements in accuracy, precision, and F1 score were up to 65.01%, 10.74%, and 40.05%, respectively. The PaDiM method demonstrated proficient classification performance within the laboratory dataset. However, its performance significantly deteriorated when applied to the production site’s dataset, which has distinct environmental conditions compared to the training dataset. To understand the rationale behind the performance improvement in the proposed two-stream network model, we examined the t-SNE plots presented in Figure 10. The feature vectors of the previous model did not exhibit clear classification boundaries for true and false samples. In contrast, the feature vectors generated by the proposed model provided the most distinct differentiation between true and false samples. The significance of this enhancement in classification features lies in its ability to alleviate the inherent bias toward true samples, which frequently possess larger datasets in comparison to false samples. The biased predictions of previous models toward true samples had a detrimental impact on precision performance, resulting in its degradation.

The two-stream network OCC model proposed in this study exhibited high classification performance with respect to both the laboratory and production site datasets. However, the validation was not sufficient for verifying the classification performance of negative samples because not enough defective samples were collected at the production site. In future studies, sufficient negative samples must be collected, and the performance of the proposed model should be further validated with those samples.

## 5. Conclusions

In this paper, we proposed a two-stream network OCC model to resolve the representation collapse problem of OCC models. The performance of the proposed model was validated in terms of the classification layer and network architecture, and comparisons were carried out using previous methods that implement image samples collected in the practical example of airbag bracket inspection. The performance results clearly indicated that the proposed model effectively addressed the representation collapse problem, resulting in enhanced inspection accuracy in comparison to existing classification models. Moreover, the classification performance of the proposed two-stream model exhibited an impressive improvement of up to 10% compared to previous classification models. This performance improvement can be accomplished using the novel two-stream network, which seamlessly integrates both general and data-specific features. The practical applications of defect inspection can greatly benefit from the implementation of the two-stream network model presented in this paper. Its incorporation is poised to make valuable contributions toward enhancing performances in vision inspection tasks.

## Figures and Tables

**Figure 1 sensors-23-05768-f001:**
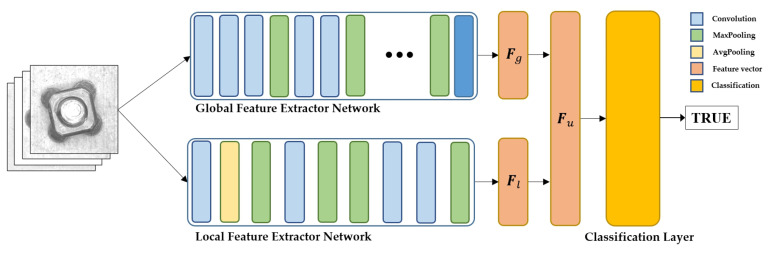
Two-stream network consisting of global and local feature extractor networks followed by a classification layer.

**Figure 2 sensors-23-05768-f002:**
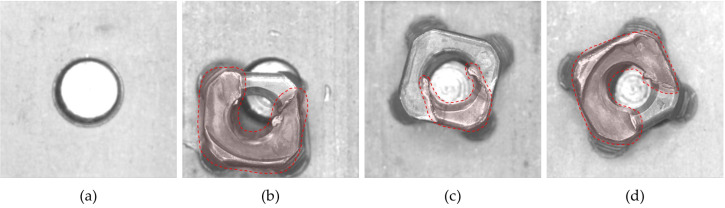
Examples of welding defects in airbag bracket inspection. (**a**) Nut omission. (**b**–**d**) Surface abnormalities.

**Figure 3 sensors-23-05768-f003:**
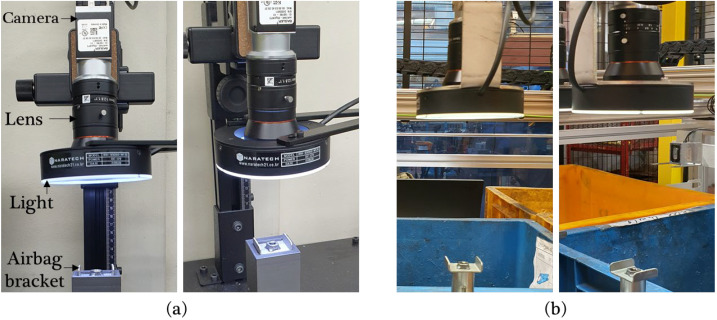
Vision system setup for collecting image samples for the model’s verification. (**a**) Vision system setup in the laboratory and (**b**) vision system setup on the production site.

**Figure 4 sensors-23-05768-f004:**
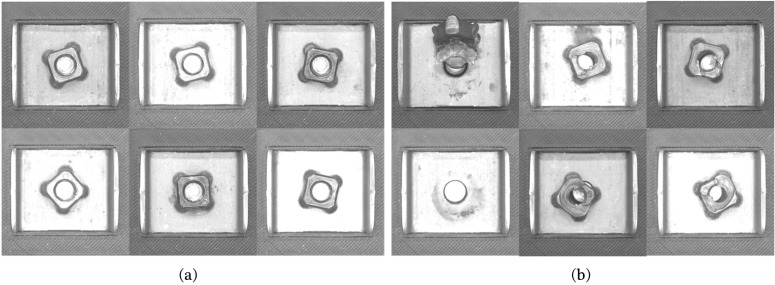
Positive and negative image samples collected in the laboratory. (**a**) Positive and (**b**) negative image samples.

**Figure 5 sensors-23-05768-f005:**
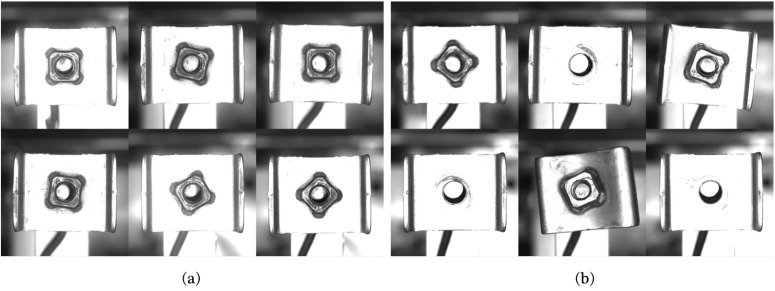
Positive and negative image samples collected at the production site. (**a**) Positive and (**b**) negative image samples.

**Figure 6 sensors-23-05768-f006:**
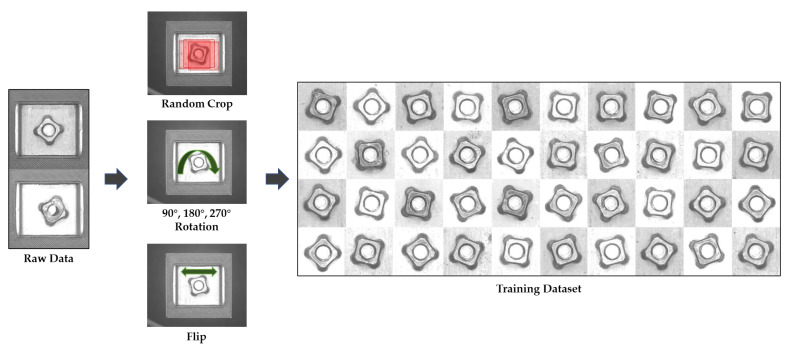
Dataset enlargement for model training by random cropping, rotating, and flipping raw images.

**Figure 7 sensors-23-05768-f007:**
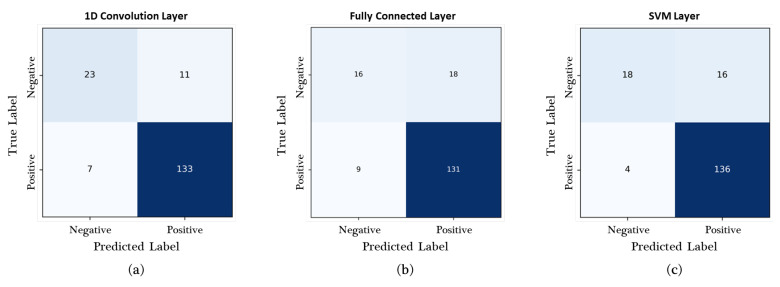
Confusion matrices of the two-stream network with three types of classification layers, as evaluated by using the laboratory dataset. (**a**) One-dimensional convolution layer, (**b**) fully connected layer, and (**c**) SVM layer.

**Figure 8 sensors-23-05768-f008:**
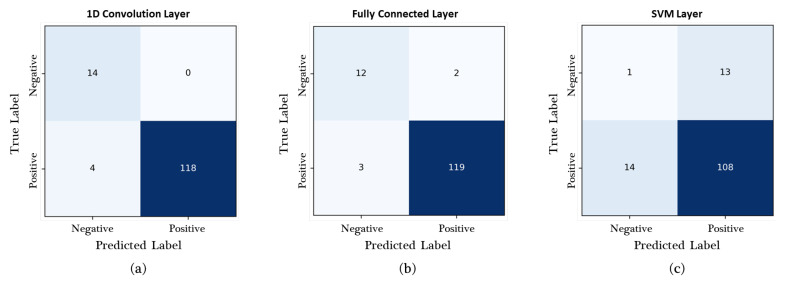
Confusion matrices of the two-stream network with three types of classification layers, as evaluated by using the production site dataset. (**a**) One-dimensional convolution layer, (**b**) fully connected layer, and (**c**) SVM layer.

**Figure 9 sensors-23-05768-f009:**
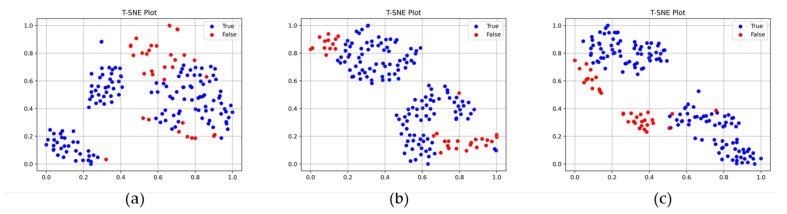
t-SNE plots of various classification models. Blue and red dots represent feature vectors of true and false samples, respectively. (**a**) Global feature extractor network, (**b**) local feature extractor network, and (**c**) two-stream network.

**Figure 10 sensors-23-05768-f010:**
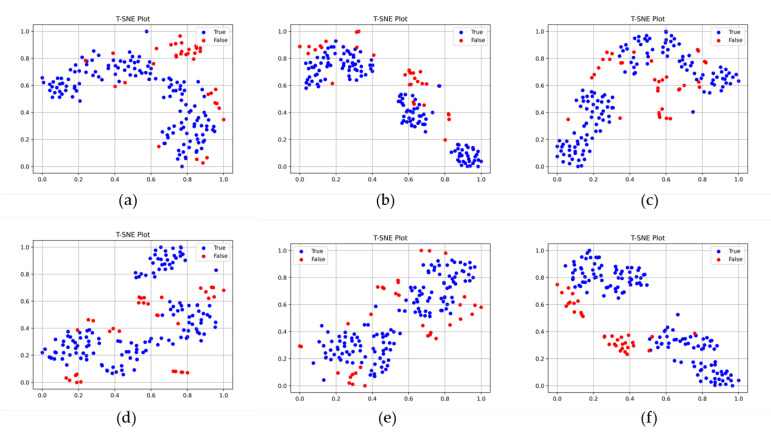
Comparison of the t-SNE plots of the previous and proposed models. (**a**) InceptionV3, (**b**) ResNet101V2, (**c**) Xception, (**d**) MobileNetV2, (**e**) VGG-16, and (**f**) proposed model.

**Table 1 sensors-23-05768-t001:** Detailed configuration of the global feature extractor network.

Type	Stride	Filter Size	Input Size
Conv2d	2	3 × 3, 32	111 × 111 × 32
Conv2d	2	3 × 3, 32	109 × 109 × 32
Conv2d	2	3 × 3, 64	109 × 109 × 64
MaxPooling2d	2	3 × 3, x	54 × 54 × 64
Conv2d	2	3 × 3, 1	52 × 52 × 80
Conv2d	2	3 × 3, 2	26 × 26 × 192
Conv2d	2	3 × 3, 1	26 × 26 × 288
3× Inception	inception module structure [41]		13 × 13 × 768
5× Inception	inception module structure [41]		6 × 6 × 1280
2× Inception	inception module structure [41]		6 × 6 × 2048
MaxPooling2d	2	1	5 × 5 × 2048

**Table 2 sensors-23-05768-t002:** Detailed configuration of the local feature extractor network.

Type	Stride	Filter Size	Output Size
Conv2d	2	7 × 7	112 × 112 × 512
MaxPooling2D	2	2 × 2	56 × 56 × 512
Conv2d	2	5 × 5	28 × 28 × 256
MaxPooling2D	2	2 × 2	14 × 14 × 256
Conv2d	1	3 × 3	12 × 12 × 128
Conv2d	1	3 × 3	10 × 10 × 128
MaxPooling2D	2	2 × 2	5 × 5 × 128

**Table 3 sensors-23-05768-t003:** Detailed configuration of three types of classification layers.

Classification Layers	Architecture			
	Type	Stride	Filter Size	Output Size
1D Convolution Layer	Covn1d	1	1 × 1	5 × 5 × 128
	Conv1d	1	1 × 1	5 × 5 × 64
	Conv1d	1	1 × 1	5 × 5 × 1
Fully Connected Layer	Dense	#of nodes: 128		
	Dense	#of nodes: 128		
	Dense	#of nodes: 1		
SVM Layer	Dense	#of nodes: 128		
	Dense	#of nodes: 128		
	Dense	#of nodes: 1		

**Table 4 sensors-23-05768-t004:** Results of performance evaluation according to the classification layer.

	Model	Accuracy	Precision	Recall	F1 Score
Laboratory dataset	1D conv.	0.8966	0.9236	0.9500	0.9366
	Fully conn.	0.8448	0.8792	0.9357	0.9066
	SVM	0.8851	0.8947	0.9714	0.9315
Production site dataset	1D conv.	0.9706	1.0000	0.9672	0.9833
	Fully conn.	0.9632	0.9835	0.9754	0.9794
	SVM	0.8015	0.8926	0.8852	0.8889

**Table 5 sensors-23-05768-t005:** Comparison of the performances of the two-stream and single-stream network models.

	Model	Accuracy	Precision	Recall	F1 Score
Laboratory dataset	Local	0.8736	0.8933	0.9571	0.9241
	Global	0.8621	0.8580	0.9929	0.9205
	Two-stream	0.8966	0.9236	0.9500	0.9366
Production sitedataset	Local	0.9310	0.9923	0.9214	0.9556
	Global	0.8971	0.9030	0.9918	0.9453
	Two-stream	0.9706	1.0000	0.9672	0.9833

**Table 6 sensors-23-05768-t006:** Performance comparison with previous models using the laboratory dataset.

Model	Accuracy	Precision	Recall	F1 Score
InceptionV3	0.8621	0.8580	0.9929	0.9205
ResNet101V2	0.8736	0.8642	1.0000	0.9272
Xception	0.8678	0.8589	1.0000	0.9241
MobileNetV2	0.8736	0.8642	1.0000	0.9272
VGG-16	0.8678	0.8589	1.0000	0.9241
PaDiM	0.8966	0.8961	0.9857	0.9388
Proposed model	0.8966	0.9236	0.9500	0.9366

**Table 7 sensors-23-05768-t007:** Performance comparison with previous models using the production site dataset.

Model	Accuracy	Precision	Recall	F1 Score
InceptionV3	0.8971	0.9030	0.9918	0.9453
ResNet101V2	0.9118	0.9104	1.0000	0.9531
Xception	0.9191	0.9173	1.0000	0.9569
MobileNetV2	0.9044	0.9037	1.0000	0.9494
VGG16	0.8971	0.9030	0.9918	0.9453
PaDiM	0.5882	1.0000	0.5410	0.7021
Proposed model	0.9706	1.0000	0.9672	0.9833

## Data Availability

Data cannot be provided due to the security reason.

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
