# Peer review of "Two-Stream Network One-Class Classification Model for Defect Inspections"

_sensors, 2023, doi:10.3390/s23125768_

Round 1

Reviewer 1 Report

The authors Seung-Hun Lee et. al  propose defect inspection method using one-class classification 13

(OCC) model to deal with imbalanced dataset. This work is meaningful and difficult. However, in the reviewer opinion the paper needs some revisions to be recommendable for publication.

1) In introduction, the research achievements in recent years should be expanded.The following papers can provide some reference for the problems.

 DOI: 10.1016/j.energy.2023.127015

2) The innovation and difficult points should be highlight.

3) The discussion and analysis should be expanded.

4) In conclusion, the important results should be expanded.

More details :

- Please check Imbalanced dataset of the machine vision systems integrating AI-based inspection algorithm.

- The author should perform more algorithm verification.

Reviewer 2 Report

The evaluation sections can be described more concisely (sections 3, 4). 

The paper describes a new approach to improve robustness of visual inspection systems. 

A serious limitation is that only one type of objects is used for evaluation. This makes it questionable if this technique is generally applicable. But I think the approach is sufficiently original to be of interest for this journal.

It is sometimes a bit wordy and tends to repeat facts. This can be streamlined.

Reviewer 3 Report

The authors discussed their major contribution of this research work as a 2-stream network with global and local feature extractor network to solve the representation collapse problem of OCC model. They verified their proposed work with automotive airbag bracket inspection. 

The overall presentation and organization of the paper is good.. except several typos and grammatical mistakes, which should be improved.

As the main contribution/claim of the novelty based on two feature extraction networks as global and local, internal layers learning representations  (like heatmaps/intermediate feature representations) should be demonstrated from these 2 branches to better understand how the concatenation of those features improving the precision/accuracy. Authors may use Keras library to generate those individual layer outputs... A comparison also must be demonstrated for corresponding layers for all other models against proposed approach. 

Also authors must discuss underfitting/overfitting problem with the dataset (may be train/val loss or accuracy plot will help)

The classification layer in Fig. 1 must demonstrate binary classification as normal and defect target classification to be consistent with input image. Authors are advised to modify this figure.

I recommend major revision of the manuscript in its present format and encourage resubmit.

Typos and grammatical mistakes must be corrected.

Reviewer 4 Report

1. There are plenty of grammatical or typos errors in the manuscript. Especially, the missing or mis-uses of definite article in the sentences.

2. Also, there are many long sentences in the manuscript. The readability of the article should be improved.

3. The numerical results show that the proposed model outperforms the existing models. For example, in Table 7, the proposed model gets 1.0000 in precision while the others only 0.90xxx. Please consider to add more descriptions on why the proposed model excels in those metrics.

4. The Conclusions Section is too short and weak to conclude the contributions and perspectives of this research.

Line 11

Recently, machine vision systems integrating AI-based inspection algorithm has ...

=> Recently, machine vision systems integrating AI-based inspection algorithms have ...

Line 12

... but practically those often suffers ...

=> ... but practically those often suffer ...

Lines 13-14

... we propose defect inspection method using one-class classification (OCC) model to deal with imbalanced dataset.

=> ... we propose a defect inspection method using the one-class classification (OCC) model to deal with an imbalanced dataset.

Line 15 

... to alleviate representation collapse problem ...

=> ... to alleviate the representation collapse problem ...

Line 23

... model ...

=> ... models ...

Line 164

... including 1D convolution layer, fully connected layer, and SVM layer ...

=> ... include a 1D convolution layer, a fully connected layer, and a SVM layer ...

Lines 348-351

Compared to the single-stream network model, the two-stream network model showed an increase up to 7.35% in accuracy, 9.70% in precision, and 3.80% in F1 score, proving that the two steam model showed better performance than the existing single-stream model.

=> Compared to the single-stream network model, the two-stream network model showed an increase up to 7.35% in accuracy, 9.70% in precision, and 3.80% in F1 score, respectively. 

Line 370

... of OCC model.

=> ... of the OCC model.

Round 2

Reviewer 3 Report

The authors addressed all the reviewer's feedbacks and concerns. The readability of the manuscript has been improved.

Authors have demonstrated the proof-of-concept towards classifying a general positive samples (one-class) and negative samples (multi-class defectivity as another class).

Authors must demonstrate the novelty of the proposed approach (towards classifying multi-class defects as (a) Nut omission. (b–d) few surface abnormalities as separate classes. Extraction and classification of global and local features of these multi-class abnormalities are quite challenging and can be proved more advantageous rather a general distinctive classification between normal and  abnormal. 

Authors are requested to compare their proposed approach with auto-encoder based approach, where even no distinct classification is required for normal and abnormal data (semi/self-supervised approach).

I recommend major revision of this manuscript in its present format.

satisfying
